# Similar hemostatic responses to hypovolemia induced by hemorrhage and lower body negative pressure reveal a hyperfibrinolytic subset of non-human primates

Morten Zaar[1], Maryanne C. Herzig[2]*, Chriselda G. Fedyk[2], Robbie K. Montgomery[2], Nicolas Prat[2,3], Bijaya K. Parida[2], Carmen Hinojosa-Laborde[1], Gary W. Muniz[1], Robert E. Shade[4], Cassondra Bauer[4¤a], Wilfred Delacruz[1¤b], James A. Bynum[2], Victor A. Convertino[1], Andrew P. Cap[2], Heather F. Pidcoke[2¤c]

**1** Center for Human Integrative Physiology, United States Army Institute of Surgical Research, Fort Sam Houston, Texas, United States of America, **2** Coagulation and Blood Research, United States Army Institute of Surgical Research, Fort Sam Houston, Texas, United States of America, **3** French Armed Forces Biomedical Research Institute (IRBA), Brétigny-sur-Orge, France, **4** Southwest National Primate Research Center, Texas Biomedical Research Institute, San Antonio, Texas, United States of America

¤a Current address: Charles River Laboratories, Ashland, Ohio, United States of America
¤b Current address: Hematology-Oncology, San Antonio Military Medical Center, Fort Sam Houston, Texas, United States of America
¤c Current address: Translational Medicine Institute, Colorado State University, Fort Collins, Colorado, United States of America
* maryanne.c.herzig.ctr@mail.mil

**Data Availability Statement:** All relevant data are within the paper and its Supporting Information files.

## Abstract

### Background

To study central hypovolemia in humans, lower body negative pressure (LBNP) is a recognized alternative to blood removal (HEM). While LBNP mimics the cardiovascular responses of HEM in baboons, similarities in hemostatic responses to LBNP and HEM remain unknown in this species.

### Methods

Thirteen anesthetized baboons were exposed to progressive hypovolemia by HEM and, four weeks later, by LBNP. Hemostatic activity was evaluated by plasma markers, thromboelastography (TEG), flow cytometry, and platelet aggregometry at baseline (BL), during and after hypovolemia.

### Results

BL values were indistinguishable for most parameters although platelet count, maximal clot strength (MA), protein C, thrombin anti-thrombin complex (TAT), thrombin activatable fibrinolysis inhibitor (TAFI) activity significantly differed between HEM and LBNP. Central hypovolemia induced by either method activated coagulation; TEG R-time decreased and MA increased during and after hypovolemia compared to BL. Platelets displayed activation by flow cytometry; platelet count and functional aggregometry were unchanged. TAFI activity

**Funding:** This study was funded by the United States Army, Medical Research and Development Command. Charles Rivers Laboratories provided support in the form of salary for author CB during the editing of this manuscript but did not have any additional role in the study design, data collection and analysis, decision to publish, or preparation of the manuscript. All other authors have no other specific funding to declare.

**Competing interests:** While author C.B. is currently employed by Charles Rivers Laboratories, this commercial affiliation does not alter our adherence to PLOS ONE policies on sharing data and materials from this current manuscript. All other authors have no competing interests.

and protein, Factors V and VIII, vWF, Proteins C and S all demonstrated hemodilution during HEM and hemoconcentration during LBNP, whereas tissue plasminogen activator (tPA), plasmin/anti-plasmin complex, and plasminogen activator inhibitor-1 did not. Fibrinolysis (TEG LY30) was unchanged by either method; however, at BL, fibrinolysis varied greatly. Post-hoc analysis separated baboons into low-lysis (LY30 <2%) or high-lysis (LY30 >2%) whose fibrinolytic state matched at both HEM and LBNP BL. In high-lysis, BL tPA and LY30 correlated strongly (r = 0.95; *P*<0.001), but this was absent in low-lysis. In low-lysis, BL TAFI activity and tPA correlated (r = 0.88; *P*<0.050), but this was absent in high-lysis.

## Conclusions

Central hypovolemia induced by either LBNP or HEM resulted in activation of coagulation; thus, LBNP is an adjunct to study hemorrhage-induced pro-coagulation in baboons. Furthermore, this study revealed a subset of baboons with baseline hyperfibrinolysis, which was strongly coupled to tPA and uncoupled from TAFI activity.

## Introduction

A deeper understanding of the effects of hemorrhage on hemostasis is central to advancing trauma care. Experimental models allow determination of pre-injury, *i.e.*, baseline values, which are unavailable in clinical trials. The intuitive technique for studying hypovolemic effects of uncontrolled bleeding is to manipulate the central blood volume by controlled bleeding, *i.e.*, blood removal (HEM) [1]. However, HEM is associated with certain risks, *e.g.*, blood may become contaminated during removal, storage, or re-infusion. Lower body negative pressure (LBNP) reduces central blood volume by sequestering blood in the legs. As such, LBNP is considered a model of bleeding because it provokes similar cardiovascular, metabolic, cerebrovascular, and immune responses to those caused by hemorrhage [2–8]. Furthermore, LBNP is safer than HEM; releasing negative pressure results in instant replenishment of the central blood volume, and the technique is non-invasive. A detailed examination of multiple levels of blood removal correlated to specific levels of LBNP revealed that LBNP closely mimicked the integrative cardiovascular response of HEM in baboons [3]. The question remained whether LBNP elicits similar hemostatic responses as HEM.

In humans, central hypovolemia elevates hemostatic activity whether it is provoked by HEM [9, 10] or by LBNP [10–13]. However, following HEM, autoresuscitation restores a portion of the intravascular plasma volume through movement of interstitial fluid into the vascular space, resulting in hemodilution [10], whereas LBNP removes a portion of the plasma volume, resulting in hemoconcentration [14]. Circulating hemostatic components, including coagulation factors and von Willebrand factor (vWF), are necessary to maintain a functional coagulation system, and the concentrations of circulating hemostatic components may determine hemostatic activity. Despite the variations in plasma volume, a direct comparison of human coagulation competence during HEM and during LBNP indicated a similar clotting response in thromboelastography-determined blood viscoelasticity [15].

The present study addresses the hypothesis that despite an expected hemodilution during HEM and hemoconcentration during LBNP, both conditions of central hypovolemia will elicit similar hemostatic responses in platelet function, coagulation, and fibrinolysis. Hemostasis is a balance between coagulation and fibrinolysis; therefore, levels of key proteins of both processes

were determined. Furthermore, how these proteins correlated with each other and with other blood components was analyzed.

## Materials and methods

The present study was performed in conjunction with a study that directly compared the cardiovascular and hormonal responses of HEM and LBNP in baboons [3]. Therefore, description of animals, anesthesia, instrumentation, and procedures for controlled and simulated bleeding are published and followed protocols approved by the Institutional Animal Care and Use Committee of the Texas Biomedical Research Institute, San Antonio, TX (Protocol number; 1287 PC 0). Furthermore, the study was conducted in compliance with the Animal Welfare Act, implementing Animal Welfare Regulations, and the principles of the *Guide for the Care and Use of Laboratory Animals* [3]. The study design is shown in S1 Fig.

### Animals, anesthesia, and instrumentation

Thirteen adult male baboons were included in the present study analysis (age: 10±1.3 y; body weight: 30.5±3.0 kg before HEM vs. 29.9±3.0 kg before LBNP, $P = 0.016$). Before instrumentation, anesthesia was induced by ketamine (10 mg kg$^{-1}$, i.m.), and maintained by ketamine (10 mg kg$^{-1}$ h$^{-1}$, i.v.) and diazepam (0.1 mg kg$^{-1}$ h$^{-1}$, i.v.). Catheters were placed in the femoral artery and in the vena cava proximal to the right atrium. Intubation maintained an open airway, which allowed for spontaneous breathing. After instrumentation, a 30–60 min supine period allowed for cardiovascular stabilization [2, 3].

### Controlled bleeding (HEM)

Blood removal from the femoral artery was performed in four steps, separated by 7 min. At each step, 6.25% of the estimated blood volume was removed by pump (50 mL min$^{-1}$), until 25% of the estimated blood volume was removed or systolic pressure was below 70 mmHg. Intermediate (MID) and maximal (MAX) blood removals were defined as 12.5% and as 25% of the estimated blood volume, respectively. Removed blood was collected in sterile blood donation bags containing citrate phosphate dextrose. After the final step of blood removal, calcium chloride (0.5 M; 1,000 × (milliliters of removed blood)$^{-1}$ mL min$^{-1}$) and removed blood (50 mL min$^{-1}$) were infused by pump via the femoral vein [2, 3].

### Simulated bleeding (LBNP)

Four weeks after HEM, the baboons were again sedated, instrumented, and subjected to LBNP as described previously [3]. The LBNP protocol was also performed in 4 steps of chamber decompression, 7–8 min at each step, until cardiovascular instability defined as systolic pressures below 70 mmHg. LBNP-intensities were determined by matching pulse pressure and/or central venous pressure observed during the previous HEM protocol. Therefore, the responses during MID and MAX LBNP corresponded with the responses MID and MAX HEM. As previously reported [3], blood loss at MID HEM was 271±26 mL (9.1±0.2 mL kg$^{-1}$) or 12.8±0.3% of the estimated total blood volume, and at MAX HEM was 516±63 mL (17.3±1.5 mL kg$^{-1}$) or 24.3±2.2% of the estimated total blood volume. MID LBNP was achieved at -41±9 mmHg, and MAX LBNP was achieved at -71±7 mmHg [3].

### Sample collection

Central venous blood was obtained during HEM and LBNP at baseline (BL), MID, and MAX time points. Post treatment blood samples (POST) were obtained 5 min after completing a 15

min blood reinfusion period for HEM or 5 min after LBNP release. Blood was collected in 3.2M citrated, EDTA, and heparin tubes for various assays. Blood samples for plasma analyses were briefly stored in wet ice before centrifugation (15 min at 2,000 $g$ and room temperature), after which, plasma was immediately isolated and stored at -80˚C until analysis [2, 3].

## Hematology

Within 30 min of collection, EDTA-stabilized blood (2 mL) was analyzed for cell counts of platelets, red and white blood cells (RBCs, and WBCs, respectively), lymphocytes, and monocytes (CELL-DYN 3700, Abbott Diagnostics).

## Thromboelastography

Coagulation competence was measured by thromboelastography (TEG Model 5000; Haemoscope Corporation). Within 30 min of collection, citrated blood was analyzed as described previously [14]. Reaction time until initial fibrin formation (R-time), rate of clot formation (α-Angle), clot elongation time (K-time), maximal amplitude reflecting clot strength (MA), and clot lysis 30 after MA (LY30) were all determined.

**Platelet activation and aggregation.** Platelet activation was measured by flow cytometry using antibodies to activated platelet glycoprotein IIb/IIIa (PAC-1) and to adhesion molecule CD62P (P-selectin). Briefly, citrated whole blood was incubated with either PAC-1 or anti-P-selectin antibodies for 15 min at room temperature. Samples were then processed in a BD FACS Lyse Wash Assistant (BD Bioscience, CA, USA) for RBC lysis, washing and fixation in 1% paraformaldehyde. Samples were analyzed on a BD FACS Canto Flow Cytometer using DIVA 6.0 software (BD Biosciences, CA, USA).

Platelet aggregation analysis used whole blood impedance aggregometry (Multiplate analyzer 5.0, Dynabyte GmbH, Germany) on blood sampled into heparin tubes [16] (LH lithium heparin Sep; BD Vacutainer). Within 30 min of sampling, platelet response to agonist at 37˚C was assessed with: collagen for primary hemostasis; adenosine diphosphate (ADPtest) for ADP-induced activation; arachidonic acid (ASPItest) for cyclooxygenase-dependent activation; thrombin receptor-activating peptide (TRAPtest) for thrombin receptor PAR-1 activation; and ristocetin (RISTO$_{high}$-test) for von Willebrand factor antigen (vWF:Ag) and glycoprotein Ib dependent activation. The analyses were performed according to the manufacturer's instructions (Instrumentation Laboratory, Munich, Germany). The area under the curve of the 6 min analysis reflects total aggregation and is expressed in arbitrary units [17].

**Plasma markers of hemostatic activity.** Fibrinogen, coagulation factors V and VIII (FV and FVIII), vWF:Ag, protein S and C activities, and D-dimer were measured using the BCS XP system (Siemens). Thrombin-antithrombin III complex (TAT), plasmin-$\alpha_2$-antiplasmin complex (PAP), tissue-type plasminogen activator (tPA) were all determined by ELISA assays (ASSERACHROM t-PA kit, Diagnostica Stago, S.A.S., France, Enzygnost PAP micro, and Enzygnost TAT micro, Siemens). Thrombin activatable fibrinolysis inhibitor (TAFI) was determined for protein content by ELISA (Imuclone Total TAFI ELISA, Sekisui Diagnostics LLC, USA) and for TAFI activity by kinetic assay (Sekisui Diagnostics GmbH, Germany). Plasminogen activator inhibitor levels (PAI-1) were determined by ELISA and measured both complexed and free PAI-1 (Zymutest PAI-1 Antigen, Hyphen BioMed, France).

**Statistical analysis.** Baseline differences between HEM and LBNP were analyzed by one-way ANOVA repeated measures, while main effects between BL, MIN, MAX, POST, HEM and LBNP were identified by two-way ANOVA repeated measures (SigmaPlot 12.0, Systat Software, San Jose, CA) and simple effects were identified by Tukey *post-hoc* test (SAS, version 9.2; SAS Institute, Cary, NC, USA). Flow cytometry data and whole blood impedance

aggregometry data with high variance were transformed by logarithm to facilitate normal distribution and variance. Multivariate analysis by Pearson correlations were performed (SAS, version 9.2; SAS Institute, Cary, NC, USA). The probability of observing chance effects that changes in the dependent variables over changing degree of hypovolemia were different from "zero" change are presented as *P*-values. A two-tailed F test for variances was performed after a Shapiro-Wilk normality test for outliers. Unless otherwise stated, data are presented as means ±SE.

## Results

### Sustained effects of blood removal

According to experimental design, a 4-week interval elapsed between HEM and LBNP trials. Despite this time interval, some hemostatic variables were affected by the previous HEM trial performed under ketamine sedation. Fig 1 compares hemostatic variables between HEM BL and LBNP BL. At LBNP BL (4 weeks after HEM BL), platelet counts (*P* = 0.003) and TEG MA (*P*<0.001) were elevated compared to HEM BL. Levels of protein C and TAT, as well as TAFI-activity were all significantly different from HEM BL values (*P* = 0.026, *P* = 0.035, and *P* = 0.042, respectively; Fig 1). Moreover, for TAT values, LBNP BL variance was significantly decreased from HEM BL (F test, *P* = 0.0063). Additionally, while a low correlation existed between RBC and platelet count at HEM BL (Fig 1; r = 0.05), a negative modest correlation was evident at LBNP BL (r = -0.476). Due to logistical constraints, these few sustained changes in coagulation parameters were deemed acceptable.

**Cell counts.** At BL and at MID hypovolemia, RBC, WBC, including lymphocyte, and monocyte counts were similar between the two trials (Fig 2 and Table 1). At HEM MAX, RBC count decreased (-5.8±5.5%, *P*<0.001*)*, whereas at LBNP MAX, the count increased (7.6 ±2.9%, *P*<0.001). These effects were sustained at POST. Overall WBC counts were similar at BL between the two trials, and increased similarly in both trials at MAX hypovolemia, (*P* = 0.027); lymphocyte count was elevated at MAX during LBNP compared to HEM (*P* = 0.005), and elevated compared to LBNP BL (*P* = 0.005). Platelet count was consistently elevated in the LBNP trial compared to the HEM trial (*P* = 0.003) for all time points, and at MAX hypovolemia compared to BL for both HEM and LBNP (*P*<0.001).

**Blood clot parameters.** Despite some differences in absolute values of parameters, HEM and LBNP resulted in similar thromboelastography responses (Fig 3). During both trials, R-time decreased at both MID and MAX (*P* = 0.067 and *P* = 0.008, respectively) hypovolemia. At BL, MA for LBNP was elevated compared to HEM (*P*<0.001); MA increased at MID (*P*<0.001 for both) and increased at MAX hypovolemia versus BL (*P*<0.001 and *P* = 0.066 for HEM and LBNP, respectively). For α-Angle, there was no significant difference between HEM and LBNP or during hypovolemia compared to BL. There was a high variance in LY30 at BL during both trials that decreased overall for both trials by MAX hypovolemia; LY30 ranged from 0% to 15% at BL and 0% to 10% at MAX hypovolemia.

**Platelet activity and function.** Platelet activity and function during both HEM- and LBNP-induced hypovolemia were mixed (Table 2). During hypovolemia, platelet activity showed a general increase by flow cytometry; however, variance increased markedly. For determination of platelet activation by both P-selectin and PAC-1, no significant difference existed at BL or during developing hypovolemia between the two trials, although HEM averaged higher values. Overall, fluorescence-positive platelets after binding of PAC-1 antibodies increased during hypovolemia (9±18%, *P* = 0.019 for PAC-1 and 5±12%, *P* = 0.08 for P-selectin). Although most baboons showed modest platelet activation by MAX hypovolemia, there was wide variance. For percentage of PAC-1 positive platelets, 2 baboons increased by 30 and

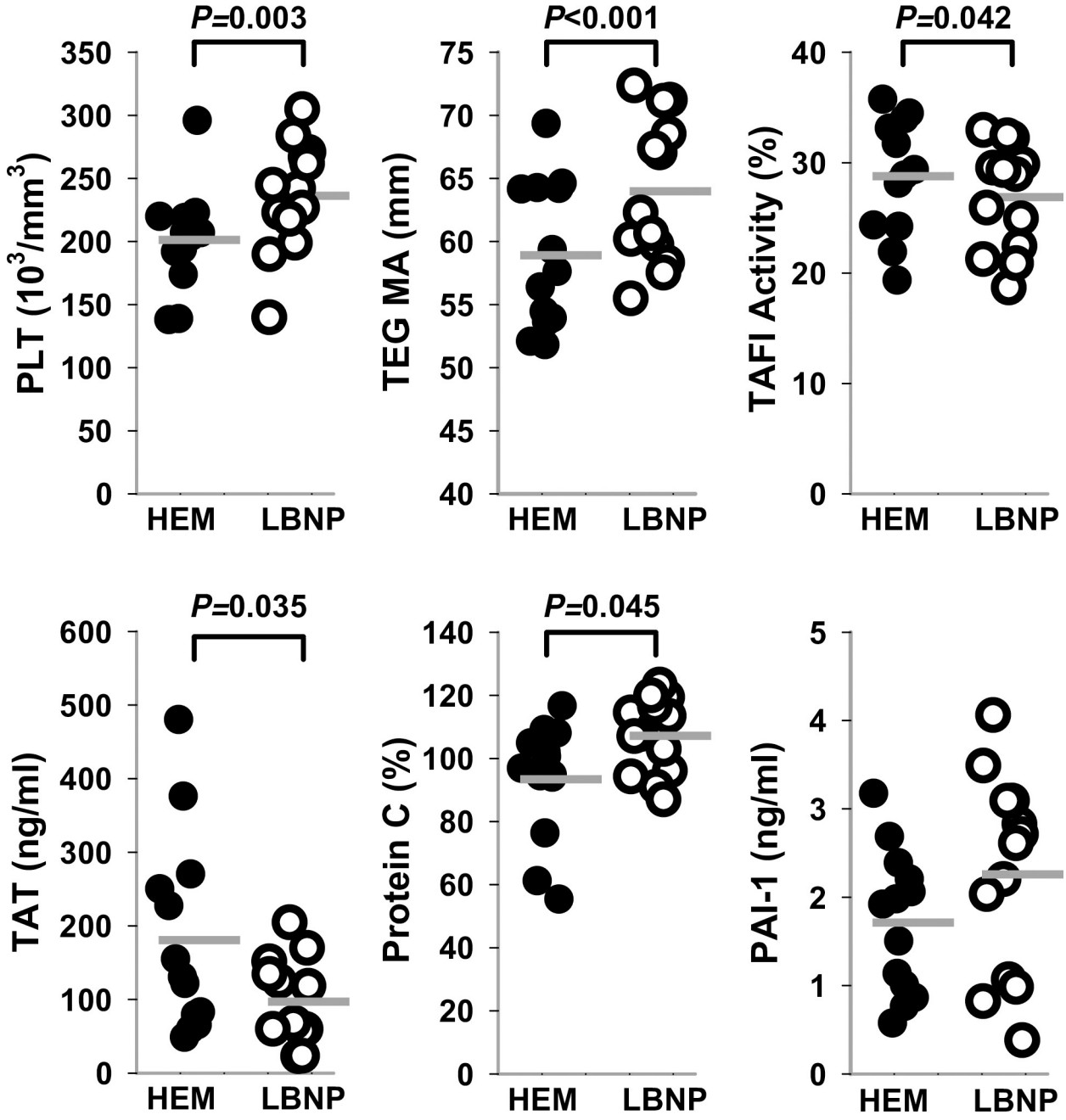

**Fig 1. Swarm plot of selected data with baseline (BL) values for blood removal (HEM) and lower body negative pressure (LBNP) trials.** LBNP BL values were obtained 4 weeks after HEM BL. *P*-values of significantly different parameters between trials as determined by one-way RMANOVA are shown above graph. Mean is presented as a gray bar.

54 points during HEM and 1 baboon increased by 70 points during LBNP. For percentage of P-selectin positive platelets, 2 baboons increased by ~20 points during HEM; one of these also increased during PAC-1; 1 baboon increased by 55 points during LBNP; this baboon had high PAC-1 during HEM. Two baboons with high PAC-1 baseline values (>21 points above median values) had high P-selectin at MAX hypovolemia. Thus, the majority of variance in both PAC-1 and P-selectin was generated by only 1/3 of the baboons.

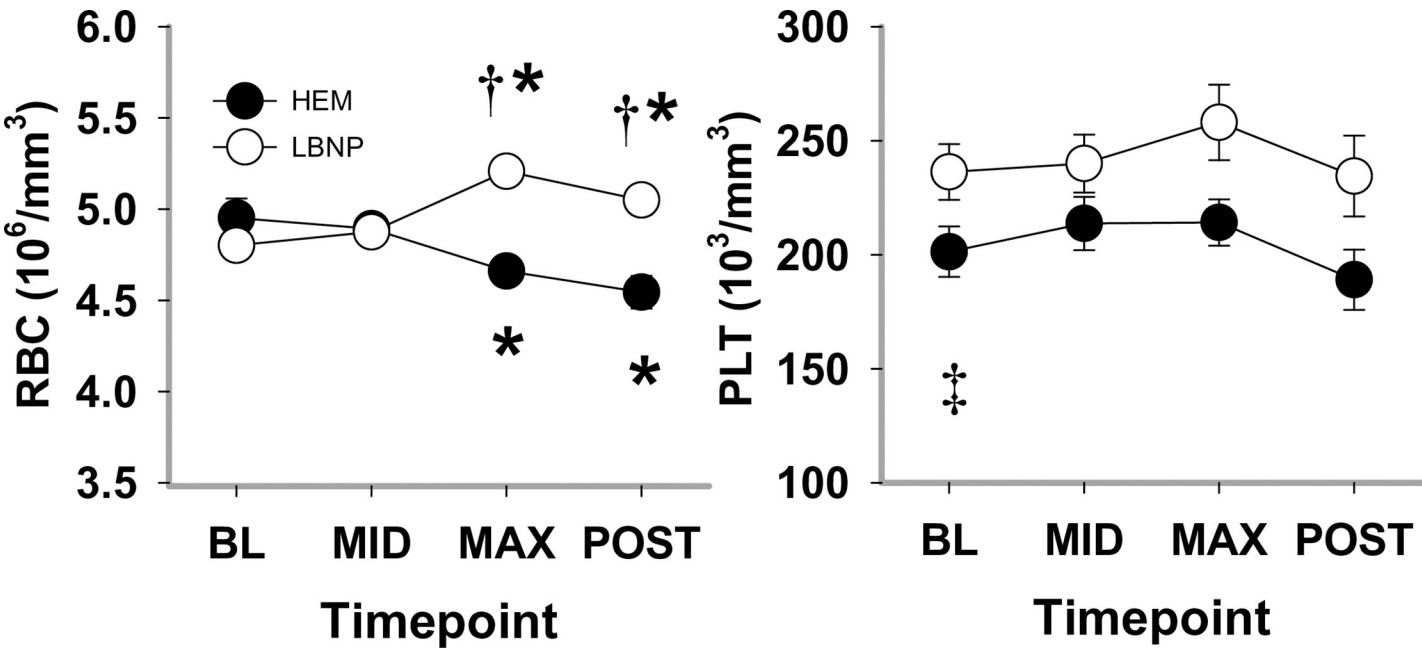

**Fig 2. Red blood cell (RBC) and platelet (PLT) counts during HEM and LBNP in anesthetized baboons.** Mean±SE, n = 13; * different from baseline ($P<0.05$); † different from blood removal ($P<0.05$); ‡ different between trials.

Platelet function tests by platelet aggregometry indicated that platelet responses to all agonists (ADP, collagen, ristocetin, TRAP, and ASPI) were unaffected by hypovolemia. However, high variance was present in all tests.

**Plasma markers of hemostasis.** In general, plasma markers of hemostasis were similar between LBNP and HEM at both BL and MID hypovolemia, and differed between the two trials at MAX and POST hypovolemia.

Pro-coagulation markers FV, FVIII, and vWF:Ag differed between LBNP and HEM by treatment ($P<0.001$, $P = 0.015$, $P<0.001$, respectively; Fig 4). FVIII activity was unchanged compared to BL during both HEM and LBNP. Both FV activity and vWF:Ag showed no significant difference between trials at BL and MID hypovolemia, while at MAX hypovolemia, FV and vWF:Ag were elevated during LBNP compared to HEM ($P = 0.013$ and $P<0.001$, respectively) and compared to its BL ($P<0.001$). The endpoint target of coagulation, fibrinogen, showed no significant increase during HEM or LBNP. Fibrinogen at POST did show a

**Table 1. Cell counts before and during blood removal (HEM) and during lower body negative pressure (LBNP).**

| | | BL | MID | MAX |
|---|---|---|---|---|
| WBC | HEM | 9.5±3.2 | 10.3±5.3 | 11.1±5.6 |
| ($10^3$ mm$^{-3}$) | LBNP | 9.0±2.4 | 9.2±3.0 | 12.4±4.7 |
| Lymphocytes | HEM | 2.5±1.2 | 2.2±0.7 | 2.2±0.9 |
| ($10^3$ mm$^{-3}$) | LBNP | 2.4±1.1 | 2.2±1.0 | 3.1±1.6 † |
| Monocytes | HEM | 0.3±0.2 | 0.4±0.1 | 0.4±0.2 |
| ($10^3$ mm$^{-3}$) | LBNP | 0.4±0.2 | 0.4±0.1 | 0.4±0.2 |

White blood cell (WBC); Mean±SD; n = 13.

† different from blood removal ($P<0.05$).

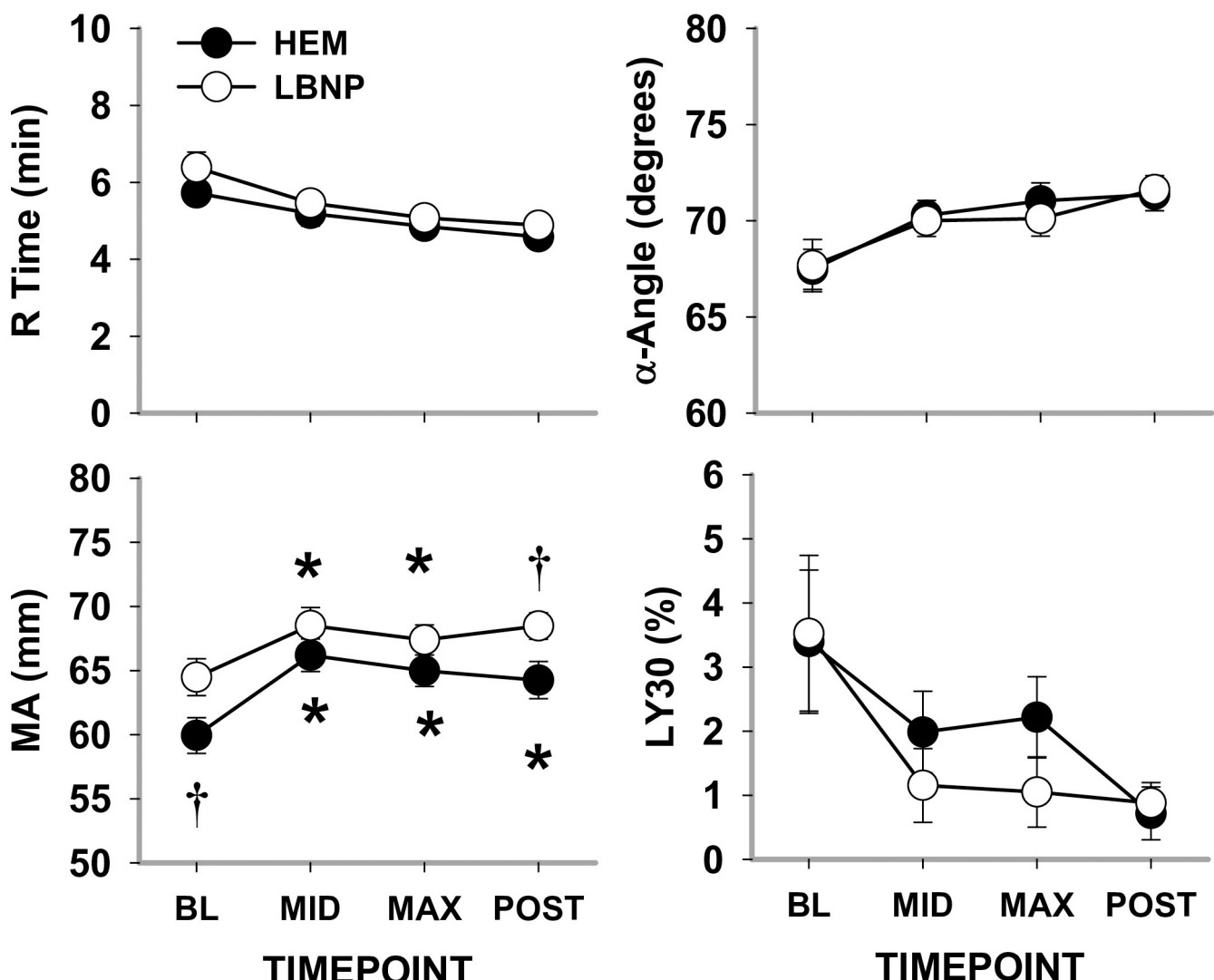

**Fig 3. Thromboelastography parameters during HEM and LBNP in anesthetized baboons.** Reaction time until initial fibrin formation (R-time), rate of clot formation (α-Angle); maximal amplitude reflecting clot strength (MA) and clot lysis at 30 min after MA (LY30). Mean ±SE, n = 13; * different from baseline ($P<0.05$); and † different from HEM ($P<0.05$).

significant decrease versus BL during HEM and a significant difference versus LBNP at this time point ($P<0.001$ and $P = 0.006$, respectively).

Anti-coagulation markers differed between the two trials during hypovolemia (Fig 5). At MAX hypovolemia, both protein S activity and protein C activity were elevated during LBNP compared to HEM ($P<0.001$), but without differing from their respective BL. During HEM hypovolemia, Protein S significantly differed from its BL at MAX ($P<0.004$). In contrast to other coagulation markers, the anti-coagulation TAT complex levels for both HEM and LBNP were significantly elevated compared to their BL at POST ($P = 0.004$). No difference between HEM and LBNP was noted for TAT complex during developing hypovolemia although at BL, MID, and POST, TAT complex had high variance.

In general, responses of fibrinolytic plasma markers were more complex. Values for both activity and protein of the fibrinolysis inhibitor TAFI were elevated by LBNP and decreased by HEM; these divergent patterns were echoed by those of hemostasis markers. In contrast,

**Table 2. Platelet activity and function before and during HEM and LBNP analyzed by flow cytometry and Multiplate.**

| | | BL | MID | MAX |
|---|---|---|---|---|
| **Flow cytometry** | | | | |
| PAC-1 | HEM | 7.0±8.6 | | 16.8±17.4* |
| (%-positive) | LBNP | 3.4±6.3 | | 13.0±23.0* |
| P-selectin | HEM | 5.1±3.3 | | 8.9±10.1* |
| (%-positive) | LBNP | 3.9±3.8 | | 9.3±15.5* |
| **Multiplate** | | | | |
| COLtest | HEM | 23±21 | 26±29 | 28±34 |
| (AUC) | LBNP | 30±27 | 23±14 | 32±21 |
| RISTO$_{high}$-test | HEM | 25±8 | 28±13 | 27±18 |
| (AUC) | LBNP | 22±11 | 26±11 | 33±17 |
| TRAPtest | HEM | 15±11 | 14±14 | 11±7.2 |
| (AUC) | LBNP | 12±11 | 11±8.9 | 14±7.9 |
| ASPItest | HEM | 2.4±2.1 | 2.4±1.9 | 2.6±1.9 |
| (AUC) | LBNP | 3.4±2.9 | 3.1±2.4 | 5.6±4.4 |
| ADPtest | HEM | 30±21 | 26±15 | 26±16 |
| (AUC) | LBNP | 27±19 | 26±21 | 34±23 |

PAC-1 antibody, collagen-induced aggregation (COLtest), von-Willebrand-factor- and glycoprotein-Ib-dependent aggregation (RISTO$_{high}$-test), thrombin-receptor-activated aggregation (TRAPtest), cyclooxygenase-dependent aggregation (ASPItest), adenosine-diphosphate-induced aggregation (ADPtest), Mean±SD, n = 9–13

* different from baseline ($P<0.05$).

fibrinolytic complexes PAI-1 and PAP as well as fibrinolysis product D-Dimer demonstrated similar patterns for LBNP and HEM with generalized increases despite high variances (Fig 6). PAI-1 assays, which measure both free and complexed PAI-1, found no difference between HEM and LBNP, but a significant difference was observed at POST vs BL ($P = 0.002$). PAP increased from BL during MAX hypovolemia by LBNP ($P<0.001$) and at HEM POST ($P = 0.001$). PAP complex averaged higher in LBNP than HEM, although there were no significant differences between the two during developing hypovolemia. Again, there was high variance as PAP complex was consistently ~100 and ~200 ng/ml for 2 baboons in the LBNP trial. Fibrinolysis activator, tPA significantly differed between HEM and LBNP at MAX hypovolemia and at POST ($P<0.001$ and $P = 0.018$, respectively). tPA increased at MAX LBNP ($P<0.001$) to between 25 and 70 ng/ml for 9 baboons and to ~150 and ~200 ng/ml for 2 baboons, resulting in the high variance, whereas at HEM MAX, tPA remained unchanged. Fibrinolysis product, D-dimer, did not differ significantly between HEM and LBNP; despite high variance, D-dimer increased between BL and MAX hypovolemia and at POST ($P = 0.037$ and $P<0.001$, respectively).

**Post-hoc analyses.** Given the wide range of LY30 values for both HEM and LBNP (Fig 7A) and the high variance of the fibrinolytic parameters (Fig 6), a post-hoc analysis was performed. The average of TEG LY30 at all time points was 1.9% with a median value of 0.5%; therefore, baboons were divided into subsets with BL LY30 <2% and BL LY30 >2%. At HEM BL, 6 baboons had LY30 <2% and 7 baboons had LY30 >2%; at LBNP BL, 6 baboons had LY30 <2% and 7 baboons had LY30 >2%. It became clear that HEM or LBNP had little effect on hyperfibinolytic baboons (Fig 7B); therefore, both LBNP and HEM data sets were combined. In order to mitigate any differences between HEM and LBNP treatments while analyzing this hyperfibrinolysis, the data were further divided into baboons whose BL LY30 matched

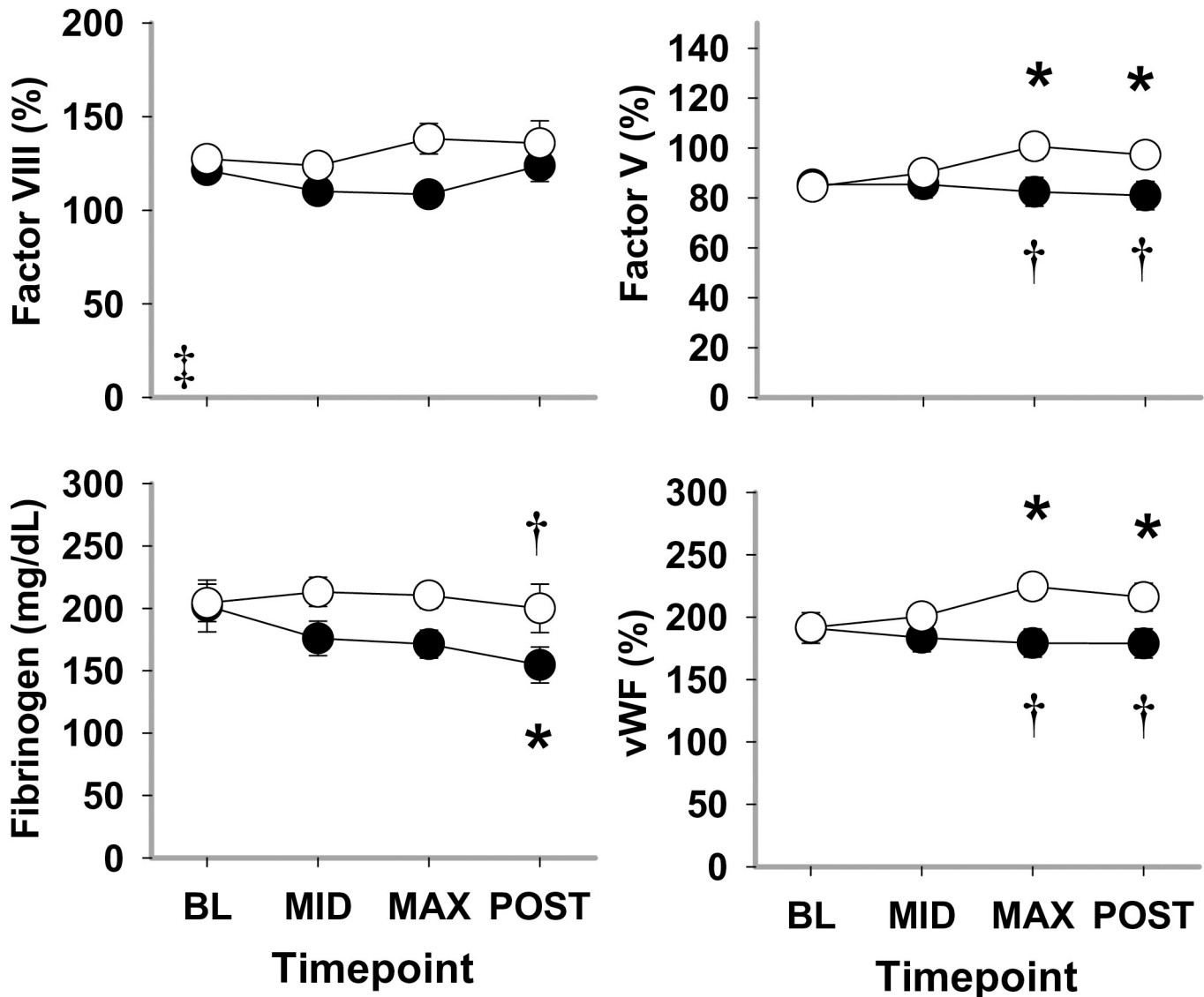

**Fig 4. Plasma pro-coagulation factors VIII and V, fibrinogen, and von Willebrand factor antigen (vWF:Ag) during HEM and LBNP in anesthetized baboons.** Mean ±SE, n = 11–13; * different from baseline ($P<0.05$); and † different from HEM ($P<0.05$).

at both HEM and LBNP; 3 baboons (n = 6 data points per time point) were designated low-lysis, while 4 baboons (n = 8 data points per time point) were designated high-lysis. The remaining baboons had both a high and a low lysis value at the two BL determinations and were not included in later analyses (Fig 7C).

Upon post-hoc analysis, it is clear that a subset of baboons are hyperfibrinolytic; the TEG measured fibrinolysis decreased during developing hypovolemia. The low-lysis baboons display a modest hyperfibrinolysis at MAX hypovolemia that is not significant with these low numbers of datapoints. High-lysis baboons had significantly less variance at BL in levels of tPA and TAFI-activity ($P = 0.001$ and $P = 0.014$, respectively) and trending less variance for TAFI-protein and TAT ($P = 0.86$ and $P = 0.15$, respectively) (Fig 8).

Pearson correlation of measured parameters was performed to explore interactions with fibrinolytic proteins (Fig 9). At BL of low-lysis baboons, no significant or trending correlations

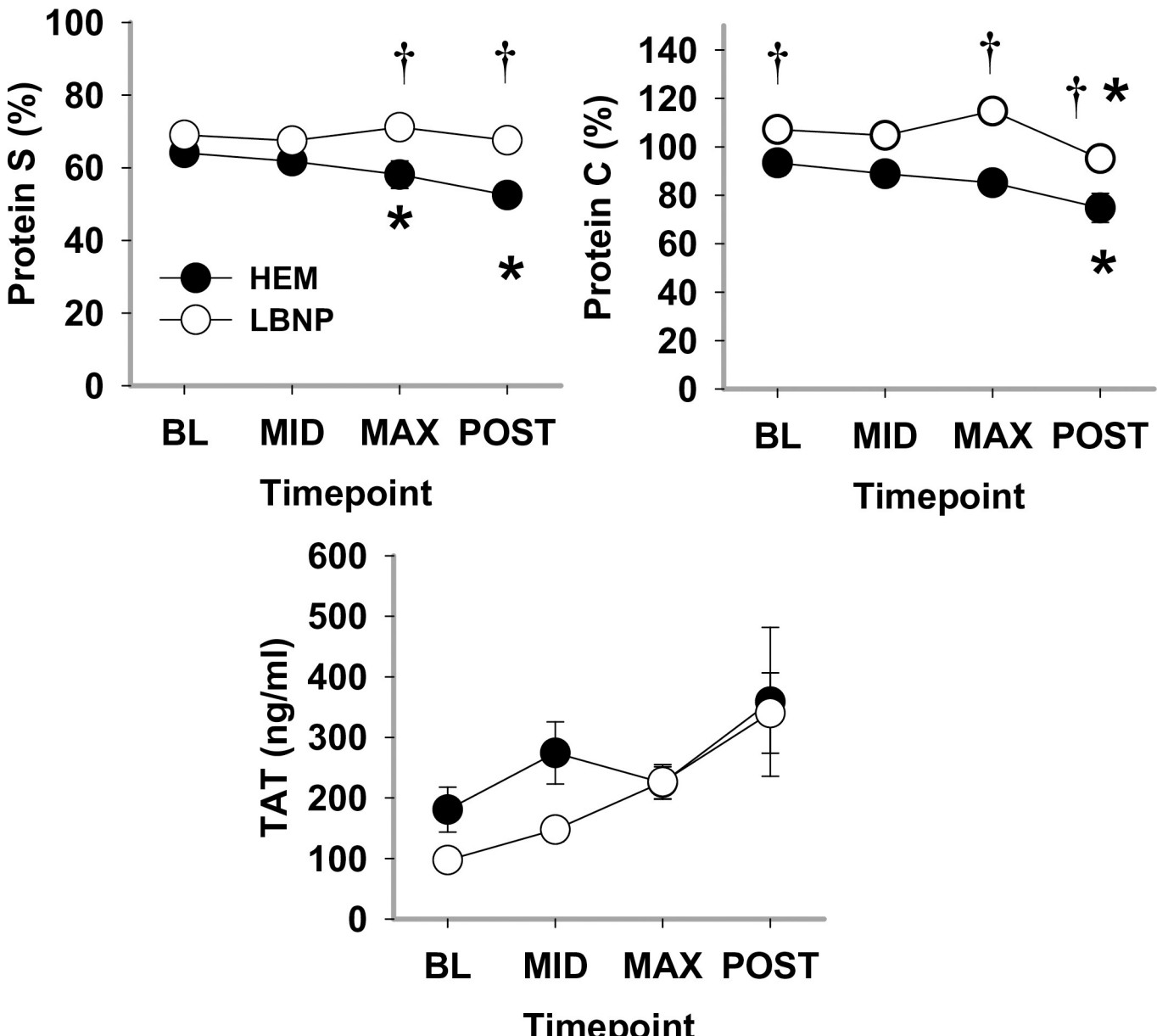

**Fig 5. Plasma anti-coagulant protein C activity, protein S activity, and thrombin-antithrombin III complex (TAT) during HEM and LBNP in anesthetized baboons.** Mean ±SE, n = 11–13; * different from baseline ($P<0.05$); and † different from blood removal ($P<0.05$).

with LY30 was observed. For high-lysis baboons, BL levels of tPA are low, but highly correlated with LY30 (r = 0.95, $P$ = 0.00029); this significant correlation is lost at MAX hypovolemia (r = -0.01, $P$ = 0.978). Both protein level and activity of TAFI were measured. A strong correlation of TAFI protein to TAFI activity was seen at BL and at MAX hypovolemia for low-lysis baboons (r = 0.92, $P$ = 0.025 and r = 0.86, $P$ = 0.061, respectively). This significant correlation is absent for high-lysis baboons at BL (r = 0.01, $P$ = 0.985) and at MAX hypovolemia (r = 0.30, $P$ = 0.558). BL TAFI activity significantly correlated to tPA protein for lowlysis baboons (r = 0.88, $P$ = 0.050); correlation was still seen at MAX hypovolemia (r = 0.79, $P$ = 0.108). No significant tPA to TAFI activity correlation is seen at BL for high-lysis baboons (r = 0.08,

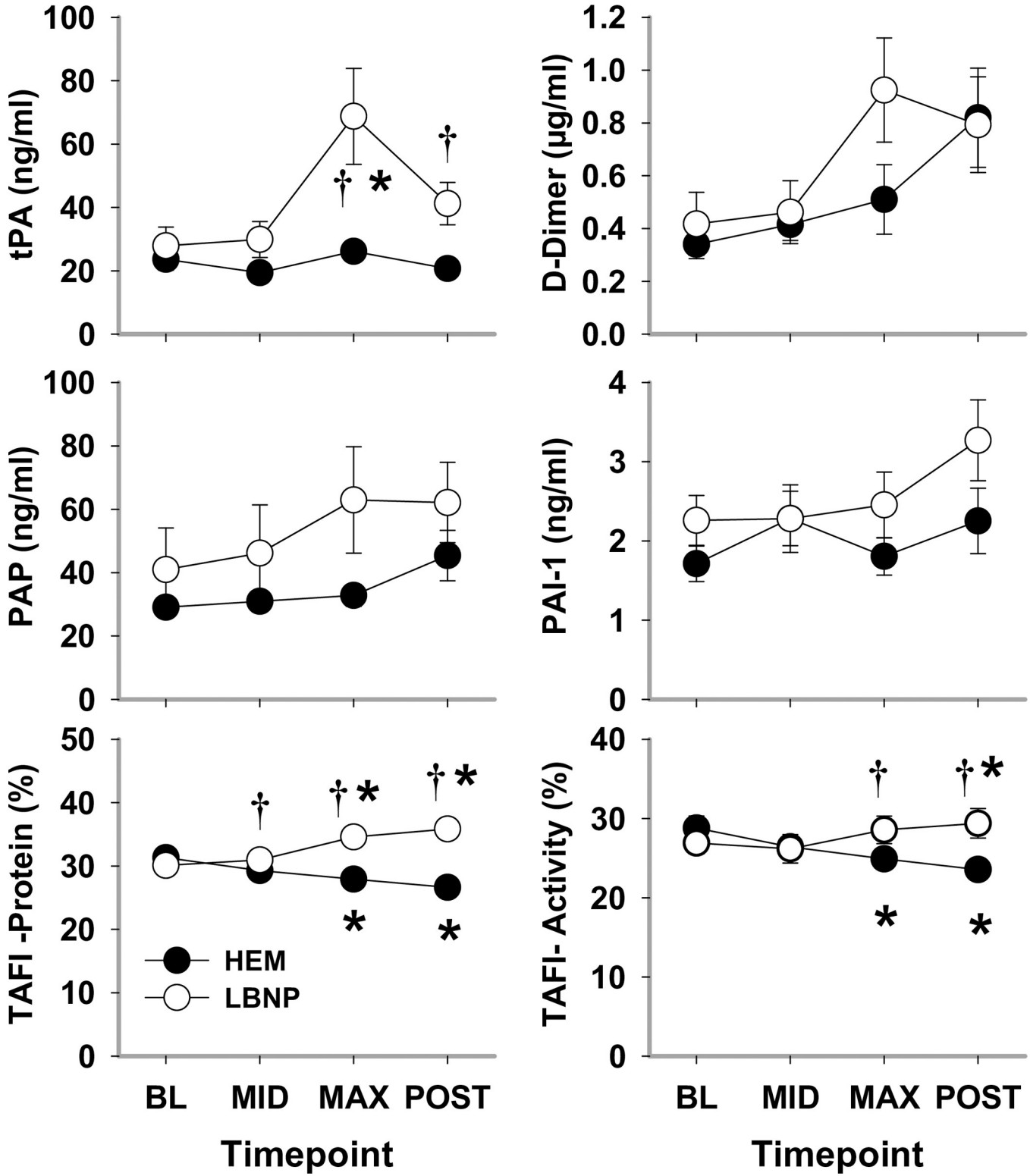

**Fig 6. Fibrinolysis indicators tissue-type plasminogen activator (tPA), D-dimer, plasmin-$\alpha_2$-antiplasmin complex (PAP), plasminogen activator inhibitor (PAI-1), thrombin activatable fibrinolysis inhibitor (TAFI) protein and activity during HEM and LBNP in anesthetized baboons.** Mean ±SE, n = 12–13; * different from baseline ($P<0.05$); and † different from blood removal ($P<0.05$).

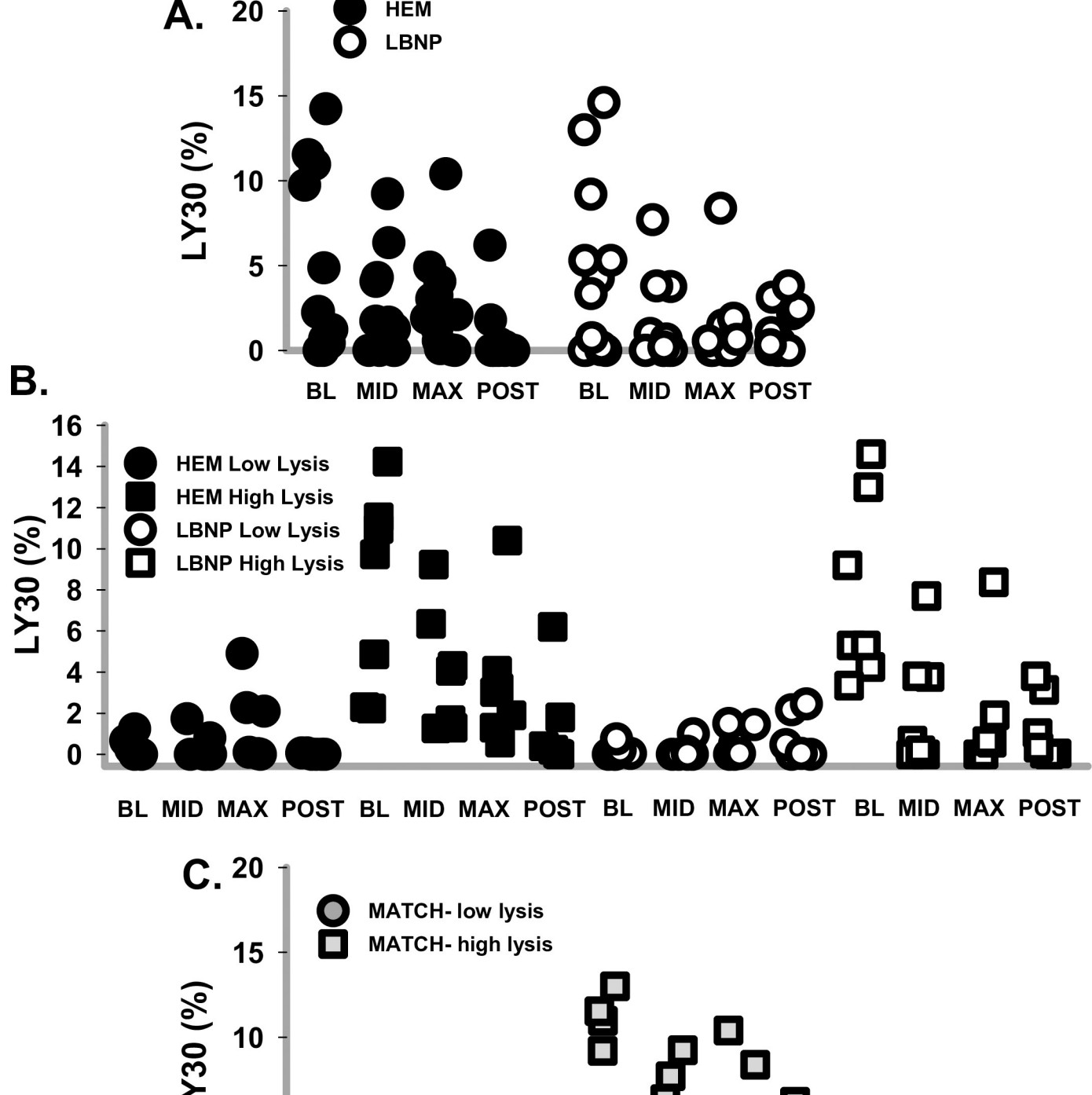

**Fig 7. Post hoc analyses of fibrinolysis data.** A. Swarm data plot shows TEG LY30 data for all subjects for HEM (dark circles) or LBNP (open circles) at each time point. B. Swarm data plot for TEG LY30 separates HEM and LBNP data sets into subjects whose $LY30_{BL} <2< LY30_{BL}$. Low-lysis designated with circles; high-lysis

designated with squares; HEM are closed symbols; LBNP are open symbols. C. Swarm data plot for TEG LY30 separated into groups whose HEM and LBNP BL matched (either both low-lysis or both high-lysis BL matched; low-lysis (LY30$_{BL}$<2) subjects designated with grey circles; BL matched high-lysis (LY30$_{BL}$>2) subjects designated with grey squares.

$P$ = 0.856); yet, a significant correlation exists with MAX hypovolemia (r = 0.88, $P$ = 0.021). At BL, the levels of fibrinolysis inhibitor PAI-1 significantly correlates with tPA for low-lysis baboons (r = 0.93, $P$ = 0.021); significance is lost at MAX hypovolemia (r = 0.47, $P$ = 0.429). The PAI-1 to tPA is loosely correlated for high-lysis baboons both at BL (r = 0.65, $P$ = 0.083) and at MAX hypovolemia (r = 0.74, $P$ = 0.094). tPA levels in high-lysis baboons at BL also significantly correlate with PAP and vWF (r = 0.73, $P$ = 0.048 and r = 0.86, $P$ = 0.0058, respectively). For high-lysis baboons, these correlations still trend at MAX hypovolemia (tPA:PAP and tPA:vWF; r = 0.74, $P$ = 0.090 and r = 0.72, $P$ = 0.107, respectively; S2 Fig). For low-lysis baboons, no correlation is seen between tPA and PAP or between tPA and vWF. PAP and vWF did show significant correlations to LY30 at BL in high-lysis baboons, while D-dimer showed a trend towards correlation (r = 0.77, $P$ = 0.026; r = 0.85, $P$ = 0.007; and r = 0.67, $P$ = 0.069, respectively; S2 Fig). Notably, high-lysis baboons show a strong negative correlation between platelets and RBCs at BL ($P$ = 0.019), which is lost at MAX hypovolemia. For low-lysis baboons, any correlation at BL or MAX hypovolemia between RBCs and platelets is absent (S2 Fig).

## Discussion

To facilitate advances in hemostasis research, it is necessary to develop experimental models that are safe, reliable, and demonstrate changes in hemostatic regulation, which correspond to pathophysiologic status. LBNP has proven to be a valid model of hemorrhage; it provokes central hypovolemia by sequestering blood in the legs, thereby resulting in similar cardiovascular and metabolic responses to the response caused by blood removal [3, 4]. As physiological responses are often integrated, the results of the present investigation extend the recognition that LBNP also appears to be a hemostatic model of hemorrhage. The major finding of this study validated our original hypothesis that LBNP in baboons generally mimic the hemostatic response caused by HEM and that the response in baboons corresponds to the human response to LBNP and HEM [3, 4]. RBCs decrease during HEM-induced hypovolemia and increase during LBNP-induced hypovolemia, as observed in humans. Platelet activity and function were mainly unaffected by hypovolemia, as demonstrated by flow cytometry and platelet aggregometry. However, both HEM and LBNP resulted in similar clotting parameters by thromboelastography; clotting initiation decreased and rate of clotting increased, while maximum amplitude also increased. These changes indicate a potentiation of coagulation during LBNP. Indications that coagulation activity actually occurred during both treatments for hypovolemia are found in the increased TAT complex levels during both trials.

As with humans, the treatments differed in effects on concentrations of some blood proteins; blood loss results in hemodilution [10], whereas LBNP results in hemoconcentration [10]. Only mild hemodilution was observed in our baboons; removing 25% of the estimated blood volume resulted in hemodilution of 5–6%, whereas removing less blood (15%) caused greater hemodilution in humans (10%) (10). This may reflect the influence of the rate of blood loss. In baboons, the blood volume was removed in 9–12 min, whereas in humans, the blood was removed in 30–40 min, allowing more time for diffusion of extravascular fluid into the circulation. Another human study reported that removing 17% of the estimated blood volume in 20–25 min resulted in a hemodilution of 2% [15]. In contrast to blood removal, 30 min of stepwise LBNP from 0 to -70 mmHg resulted in hemoconcentration of 5–6% in baboons. Again,

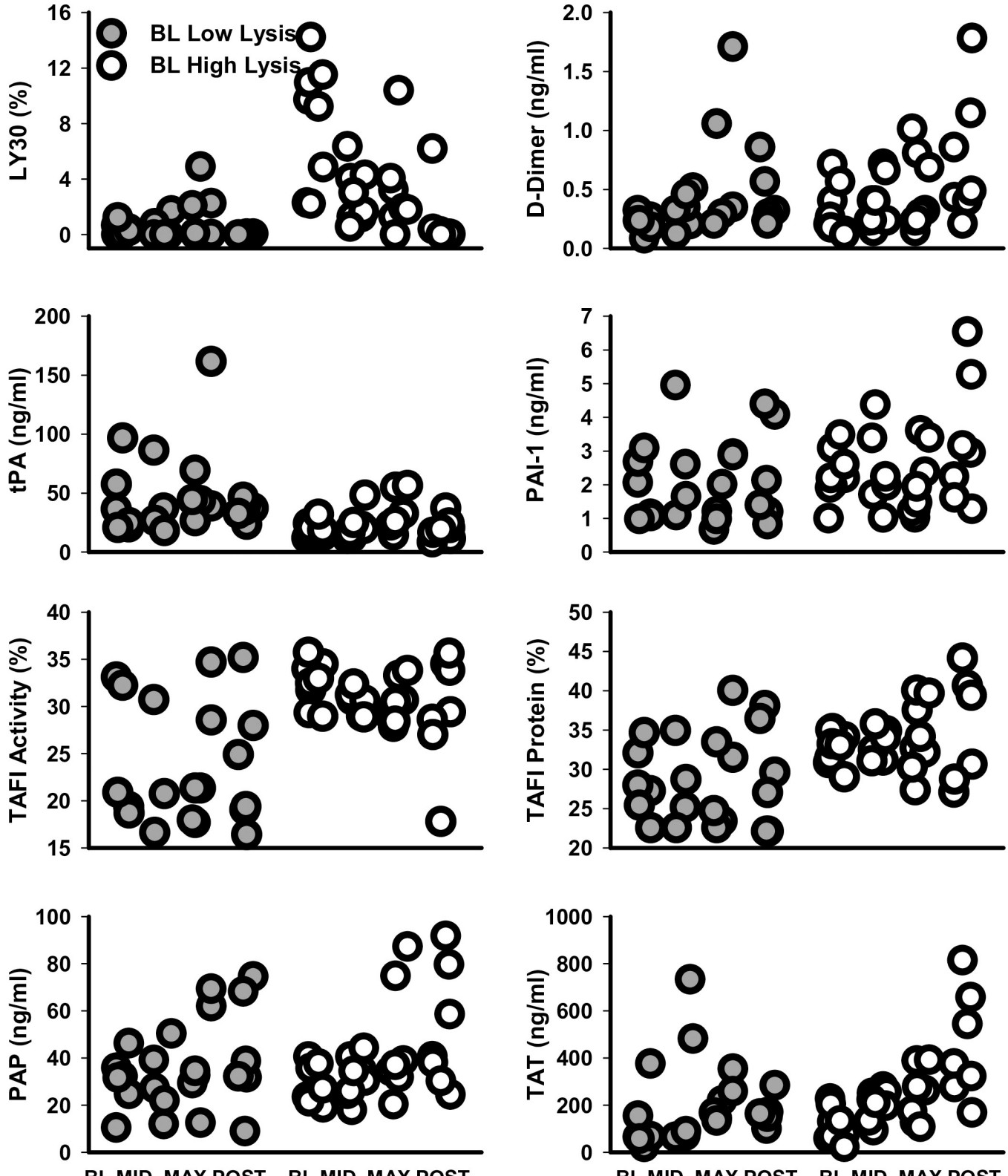

**Fig 8. Post Hoc analysis of fibrinolytic indicators grouped low-lysis (LY30$_{BL}$ <2%) or high-lysis (LY30$_{BL}$ >2%) baboons whose fibrinolytic state matched at both HEM and LBNP BL.** Swarm plots show all data points for fibrinolytic indicators LY30, D-dimer, tPA, TAFI-activity, TAFI-protein and PAP as well as the indicator of coagulation, TAT during HEM and LBNP treatment. Solid circles are BL low lysis; open circles are BL high lysis.

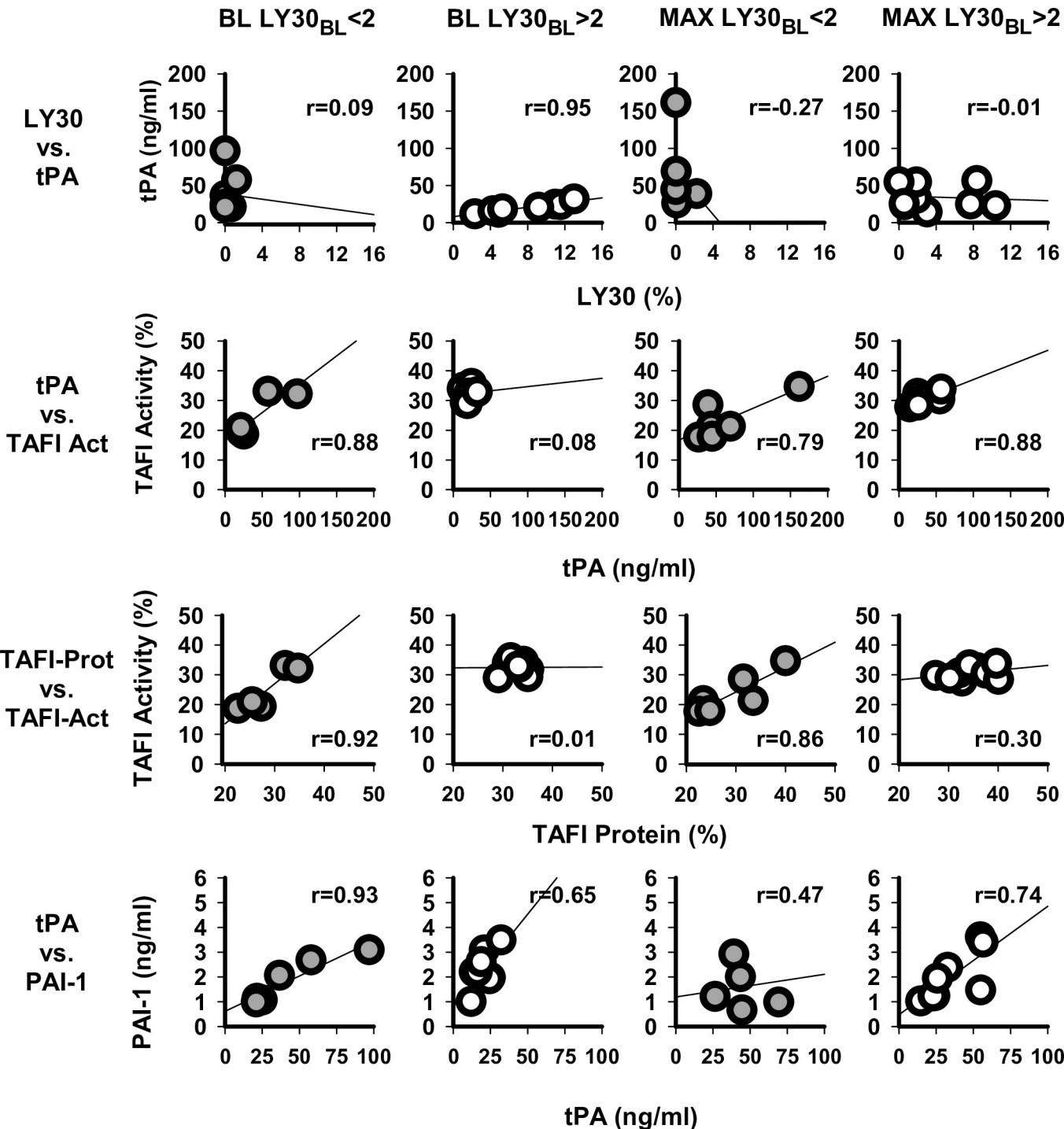

**Fig 9. Post hoc analysis of correlations of LY30 to tPA; tPA to TAFI-activity; TAFI-protein to TAFI-activity; and tPA to PAI-1.** Graphs show all data points at individual time points BL and MAX hypovolemia for low-lysis and high-lysis groups. Solid circles are LY30$_{BL}$ <2%; open circles are LY30$_{BL}$ >2%.

this is a milder hemoconcentration than is observed in humans, where 20 min of stepwise LBNP from 0 to -65 mmHg results in hemoconcentration of 10% [10]. Although hemoconcentration vs. hemodilution of hemostatic proteins might have been expected to impact

thrombogenesis [18], this was not seen by thromboelastography. In general, pro-coagulant proteins followed the hemodilution by HEM (decreased concentration with MAX hypovolemia) and hemoconcentration by LBNP (increased concentration with MAX hypovolemia). Responses of anti-coagulant proteins were more complex; protein C was already significantly elevated at LBNP BL compared to HEM BL, while TAT complexes increased in both groups. As such, our results are the first to demonstrate that the hemostatic response to hemorrhage appears to be independent of hemodilution since it also occurs with hemoconcentration, but instead appears to be primarily stimulated by central hypovolemia that is common in both HEM and LBNP.

In comparing baseline levels of hemostatic parameters between the two trials, the increased levels of platelets and protein C, the increased TEG MA, the decreased TAFI-activity and TAT complexes ($P<0.05$) suggest a sustained response of the baboons to the initial blood removal under ketamine sedation 4 weeks previously. Additionally, the variance of several proteins, including TAT, PAP and tPA, decreased by LBNP BL. The previously uncorrelated levels of RBCs and platelets at HEM BL (r = 0.0409; r = 0.779), is trending towards correlation at LBNP BL (r = -0.293; $P$ = 0.0906). Whether this is a hemodilution effect of the introduction of the anti-coagulant with the blood reinfusion, a citrate effect of anti-coagulant, an effect of sedation or a hemostatic response to rapid blood loss despite re-transfusion is unknown. Ketamine has a known effect of increasing platelets 15 days after infusion [19], why this would extend to hemostatic proteins such as Protein C, TAT or TAFI activity, all related to thrombin regulation, is unclear. During the 4-week recovery period between the two trials, the mass of the baboons decreased by 2%. We were unable to identify a specific cause of the decrease in mass, *e.g.*, loss of muscle mass, fat, or blood volume. In contrast, most other parameters were unchanged, *i.e.*, RBC, WBC, monocytes, lymphocytes, granulocytes, hematocrit, pH, platelet aggregometry responses, flow cytometry analyses, coagulation protein levels of factor VIII, vWF, fibrinolytic indicators, including PAI-1 and TAFI-protein, D-dimer, and TEG parameters, *i.e.*, rate of initiation and rate of clot formation. With the bulk of the BL responses unchanged, it seems unlikely that the few BL changes observed influenced the responses to progressive central hypovolemia reported in this study. However, it is important to recognize that despite blood replacement, some effects of a simulated hemorrhage were sustained 4 weeks later.

Fibrinolysis is controlled both by the conversion of plasminogen to plasmin as plasmin converts fibrin clots into fibrin degeneration products (*e.g.*, D-dimer), and by the neutralization of plasmin. Conversion of plasminogen to plasmin is facilitated by tPA, whereas PAP complex is an indicator of plasmin neutralization by $\alpha_2$-antiplasmin. Fibrinolysis can be inhibited by direct interaction of tPA with plasminogen activator inhibitor, PAI-1, to form inactive complexes. Additionally, thrombin activatable fibrinolysis inhibitor, TAFI, will cleave binding sites on fibrin, preventing tPA binding and activation of plasminogen. Multiple metabolic pathways intersect. Bradykinin can stimulate tPA release but it is itself degraded by plasma renin activity in the conversion of angiotensinogen. Endothelial cells release tPA during adrenergic stimulation [20], while $\alpha_2$-antiplasmin is contained in $\alpha$-granules of platelets [21]. Our data show that in a subset of baboons, tPA appears to be dysregulated from inhibition in the baseline condition; thus, tPA is active and capable of promoting fibrinolysis despite low levels of tPA.

In humans, both HEM and LBNP elicit a developing hyperfibrinolysis by TEG [14]. In baboons, the expected hyperfibrinolysis events were not immediately evident. High variance was present but both D-dimers and PAP levels trended towards an increase during developing hypovolemia. However, the LY30 values in thromboelastography were surprising. Not only was a hyperfibrinolysis present at baseline, it had high variance and was independent of HEM or LBNP (Fig 3). Values ranged from 0% lysis to 14.6%, averaging 4.4% ±5.0%. While the

overall average of LY30 for all time points and all trials was 1.9%, the median LY30 was 0.5%. This hyperfibrinolysis of TEG LY30 values largely resolved during hypovolemia; by POST, values for LY30 range from 0% to 6.2% lysis with an average of 0.9%. Thus, a post-hoc analysis was initiated to dissect possible causes of this variant BL hyperfibrinolysis. Baboons were sorted into low-lysis at BL (LY30$_{BL}$ <2%) and high-lysis at BL (LY30$_{BL}$ >2%). To further analyze and mitigate possible divergent effects of LBNP and HEM, only baboons whose LY30$_{BL}$ matched were grouped together. This post-hoc analysis revealed a hyperfibrinolytic subset of baboons whose baseline *in vitro* fibrinolysis in thrombolastography decreased over the course of central hypovolemia. In contrast, the low-lysis subset of baboons trended towards a modest increase in *in vitro* fibrinolysis at MAX hypovolemia.

It is worth emphasizing that thromboelastography is an *in vitro* assay that does not necessarily recapitulate *in vivo* blood hemostatic changes, but instead reflects the potential responses of blood to *in vivo* stimulation. Thus, the paradoxical data of plasma D-dimer, PAP, and TAT levels that are all lowest at BL despite *in vitro* hyperfibrinolysis suggests that rather than an ongoing thrombotic event with hyperfibrinolysis, high-lysis baboons are primed to respond to potential thrombotic events with high fibrinolysis. As hypovolemia develops and actual thrombotic events occur (a trend towards elevated TAT, PAP, and D-dimer), the system responds with activation of coagulation factors (suggested by increased TEG R-time, α-Angle, and MA) and fibrinolytic activation (PAI-1, and TAFI-protein and activity). Understanding that correlation is not causation, it is nonetheless compelling that tPA is lowest at BL in high-lysis baboons and, yet, is tightly correlated with LY30. At BL in low-lysis baboons, TAFI-activity is tightly correlated with TAFI-protein and with tPA protein; this correlation is not present in high-lysis baboons.

The variation in fibrinolysis may mimic what is seen in humans. Variation in fibrinolysis in humans was found critical in survival of trauma. Both fibrinolysis shutdown and the converse, hyperfibrinolysis, were associated with increased mortality [22]. In the hyperfibrinolytic patient, tPA release was identified as the cause of hyperfibrinolysis [23, 24]. In the present study, dysregulation of tPA again appears to be the cause of hyperfibrinolysis. The highest BL lysis is seen with the lowest tPA levels. The loss of tPA correlation with TAFI-activity in the high-lysis subset might be indicative of a general dysregulation for TAFI in this group.

## Limitations

In order to match the magnitude of reduced central blood volume with the two experimental conditions using the measurement of central venous pressure, all subjects were subjected to LBNP 4 weeks after HEM. The inability to randomize order of experimental condition (*i.e.*, LBNP and HEM) increased the possibility of an order effect as reflected by sustained differences from BL in a few parameters including platelets. While differences between HEM and LBNP hemostatic responses could be explained by an order effect, no such differences were observed. It is unlikely that the similarities in hemostatic responses to central hypovolemia caused by either LBNP or HEM could be explained by the absence of random order. In the analysis of hyperfibrinolysis, it is important to remember that correlations, while compelling, are not causation. Also, in the post-hoc analysis, data points were reduced to 6 and 8 data points per group, limiting the interpretation of statistical analyses and exacerbating random error.

## Conclusion

In anesthetized baboons, LBNP was validated as a hemostatic model of bleeding dependent upon both severity of hypovolemia and type of hemostatic analysis. At MID hypovolemia,

where LBNP and HEM did not affect cell counts and plasma markers, and at MAX hypovolemia where LBNP caused a mild hemoconcentration and HEM caused a mild hemodilution, thromboelastography elicited responses to potential thrombotic events that were nearly identical. Therefore, if evaluating hemostasis by blood viscosity, LBNP elicits similar responses to HEM. If analysis relies solely on plasma markers, the LBNP-induced hemoconcentration at MAX hypovolemia may introduce a complication to interpretation of results. This study also suggested that blood removal by HEM results in sustained changes that prevail at least 4 weeks post simulation. Finally, this study revealed a subset of baboons in this non-human primate model which are hyperfibrinolytic. Their baseline response to a thrombotic event is poised for hyperfibrinolysis; tPA appears uncoupled from TAFI-activity. Thus, the baboon is revealed as an excellent model for LBNP studies and for analyzing fibrinolysis in response to trauma.

## Supporting information

**S1 File. Animal methods.**
(DOCX)

**S1 Fig. Test design and study protocol.**
(TIF)

**S2 Fig. Post hoc analysis of correlations for baboons grouped as low-lysis (LY30 <2%) or high-lysis (LY30 >2) for tissue-type plasminogen activator (tPA) vs. plasmin-α2-antiplasmin complex (PAP) and vs. von Willebrand factor (vWFclot lysis at 30 min (LY30) vs. D-dimer; and red blood cell count (RBC) vs. platelet count (PLT) at both BL and MAX hypovolemia.**
(TIF)

## Acknowledgments

We thank Jessie D. Fernandez for her contribution in preparing this paper.

## Disclosures

The opinions or assertions contained herein are the private views of the authors, and are not to be construed as official or as reflecting the views of the Department of the Army or the Department of Defense.

## Author Contributions

**Conceptualization:** Carmen Hinojosa-Laborde, Robert E. Shade, Victor A. Convertino, Andrew P. Cap, Heather F. Pidcoke.

**Formal analysis:** Morten Zaar, Maryanne C. Herzig, Chriselda G. Fedyk, Robbie K. Montgomery, Nicolas Prat, Bijaya K. Parida, Carmen Hinojosa-Laborde, Heather F. Pidcoke.

**Funding acquisition:** Victor A. Convertino, Andrew P. Cap, Heather F. Pidcoke.

**Investigation:** Maryanne C. Herzig, Chriselda G. Fedyk, Robbie K. Montgomery, Nicolas Prat, Bijaya K. Parida, Carmen Hinojosa-Laborde, Gary W. Muniz, Robert E. Shade, Cassondra Bauer, Wilfred Delacruz, Victor A. Convertino, Heather F. Pidcoke.

**Methodology:** Carmen Hinojosa-Laborde, Gary W. Muniz, Robert E. Shade, Cassondra Bauer, Victor A. Convertino, Andrew P. Cap, Heather F. Pidcoke.

**Project administration:** Chriselda G. Fedyk, Carmen Hinojosa-Laborde, Gary W. Muniz, Robert E. Shade, Victor A. Convertino, Heather F. Pidcoke.

**Resources:** Gary W. Muniz, Robert E. Shade, Cassondra Bauer, Victor A. Convertino, Andrew P. Cap.

**Supervision:** Chriselda G. Fedyk, Carmen Hinojosa-Laborde, Robert E. Shade, James A. Bynum, Andrew P. Cap, Heather F. Pidcoke.

**Writing – original draft:** Morten Zaar, Maryanne C. Herzig.

**Writing – review & editing:** Morten Zaar, Maryanne C. Herzig, Chriselda G. Fedyk, Robbie K. Montgomery, Nicolas Prat, Bijaya K. Parida, Carmen Hinojosa-Laborde, Gary W. Muniz, Robert E. Shade, Cassondra Bauer, Wilfred Delacruz, James A. Bynum, Victor A. Convertino, Andrew P. Cap, Heather F. Pidcoke.

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
