## [Decision Letter · Decision Letter 0]

20 Apr 2020

PONE-D-19-35990

Similar hemostatic responses to hypovolemia induced by hemorrhage and lower body negative pressure reveal a hyperfibrinolytic subset of non-human primate

PLOS ONE

Dear Dr Maryanne Herzig,

Thank you for submitting your manuscript to PLOS ONE. After careful consideration, we feel that it has merit but does not fully meet PLOS ONE’s publication criteria as it currently stands. Therefore, we invite you to submit a revised version of the manuscript that addresses the points raised during the review process.

We would appreciate receiving your revised manuscript by May 04 2020 11:59PM. To enhance the reproducibility of your results, we recommend that if applicable you deposit your laboratory protocols in protocols.io, where a protocol can be assigned its own identifier (DOI) such that it can be cited independently in the future. For instructions see: http://journals.plos.org/plosone/s/submission-guidelines#loc-laboratory-protocols

We look forward to receiving your revised manuscript.

Kind regards,

Nandu Goswami, MBBS, PhD

Academic Editor

PLOS ONE

Journal Requirements:

2. To comply with PLOS ONE submission guidelines for non-human primate studies (http://journals.plos.org/plosone/s/submission-guidelines#loc-animal-research), we request that you please provide the following information in the Methods section of your manuscript:

* Please provide details explaining the surgeries performed in this study. Thank you for providing a reference, we ask that you please include these details in this manuscript as well in the methods section or as supporting information.

*Please describe the post-operative care received by the animals, including the frequency of monitoring and details of how you minimized animal suffering and distress.

* Details of animal welfare (e.g., shelter, food, water, environmental enrichment)

*The source of the animals used in this study, where they were bred and housed prior to this study.

*Please explain the fate of the animals used in this study. If they were euthanized, please describe the method of sacrifice.

Thank you for your attention to these requests, we look forward to hearing from you.

We note that one or more of the authors are employed by a commercial company: Charles River Laboratories

Reviewers' comments:

Reviewer's Responses to Questions

**Comments to the Author**

1. Is the manuscript technically sound, and do the data support the conclusions?

Reviewer #1: Yes

Reviewer #2: Yes

2. Has the statistical analysis been performed appropriately and rigorously? 

Reviewer #1: I Don't Know

Reviewer #2: Yes

3. Have the authors made all data underlying the findings in their manuscript fully available?

Reviewer #1: Yes

Reviewer #2: Yes

4. Is the manuscript presented in an intelligible fashion and written in standard English?

Reviewer #1: Yes

Reviewer #2: Yes

5. Review Comments to the Author

Reviewer #1: The paper addresses an important research question. The main finding of this study is that LBNP (lower body negative pressure) causes the same hemostatic responses as HEM (controlled blood removal) in baboons. HEM causes decreased RBCs but LBNP causes increased RBCs. Both HEM and LBNP did not affect platelet activity and function as evaluated by means of aggregmetry measurements. Thrombelastometry measurements and increased TAT complex levels revealed that both HEM and LBNP cause a shift towards hypercoagulability. The hemostatic response to hemorrhage appears to be primarily stimulated by central hypovolemia and not by hemoconcentration (LBNP) or by hemodilution(HEM). Both HEM and LBNP did not immediately cause hyperfibrinolysis.

I have some global and minor concerns that should be addressed before this paper can be accepted. There is a need to provide new references and to expand the limitations section.

Global comments

It is well established that HEM causes hemodilution and LBNP causes hemoconcentration. In the present study it is shown, however, that TAFI activity and protein demonstrates hemodilution during LBNP and hemoconcentration during HEM. The authors should discuss on this discrepancy.

Sustained response of baboons to blood removeal 4 weeks previously was observed. Authors should discuss on possible physiological/biochemical reasons.

The authors state that blood samples for plasma analyses were briefly stored in wet ice before centrifugation (lines 138-139). However, storage at low temperatures can cause platelet activation. Can authors exclude that platelet activation occurred in their study? The authors state that platelet aggregation measurements were performed by using heparin tubes (lines 160-161). However, platelet aggregation measurement are usually performed in citrated blood samples (see Cvirn et al. (2012) Coagulation changes during presyncope and recovery. PLOS ONE 7 (8), e42221). Can authors comment on this?

Specific comments

1. The title is a bit misleading. The study was not designed to identify or analyze the hyperfibrinolysis subset of non - human primates. Revision of the title should be considered, e.g., „Similarities in hemostatic response to LBNP and controlled hemorrhage in baboons – a crossover study“.

2. Line 70/ Line 72: a newer reference about LBNP should be included (Goswami et al. 2018. Lower body negative pressure: Physiological effects, applications, and implementation. Physiol Rev. 2019; 99(1): 807-851)

3. Line 77: Newer references about LBNP and coagulation should be included (Cvirn et al. 2019 Coagulation changes induced by lower-body negative pressure in men and women. J Appl Physiol (1985). 2019; 126(5):1214-1222).

4. There is no indication of the study design.

5. Sample size: Were sample size calculations done? If yes, on what basis?

6. In the study is acknowledged that the carry-over effect exists that extends over the washout period; however, it is incorrect to ignore the design of the study and just perform a comparison of treatments (test conditions). The possibility of a period effect should be tested – to compare the differences between the periods in the group. Additionally, the possibility of the treatment period interaction also exists (see: Altman DG. Clinical trials. In: Practical statistics for medical research. London: Chapman & Hall; 1999. p. 467–71). In the study, that would not be possible, since subject allocation to one or the other test condition (HEM or LBNP) was not random (randomization was not achieved). The authors should discuss this.

7. The limitations sections should be expanded so that it takes into account the above outlined concerns.

8. Lines 205-206: It is stated here that a low correlation existed between RBC and platelet count at HEM BL (Fig. 1; r=0.05). This correlation, however, is not presented in Fig. 1.

Reviewer #2: The authors describe, characterize and compare hemostatic profiles of two techniques of modeling hypovolemia in non-human primates; either hemorrhage or lower body negative pressure. The authors have a well written manuscript. the coagulation response of each technique provides important information to researchers in this area. Studies using non-human primates are essential to promote improvements in care for the injured patient. The authors findings are novel and robust. There conclusions are appropriate and limitations well described.

6. PLOS authors have the option to publish the peer review history of their article (what does this mean?). If published, this will include your full peer review and any attached files.

Reviewer #1: No

Reviewer #2: No

---

## [Author Response · Author response to Decision Letter 0]

4 May 2020

Dear Dr. Goswami,

We are pleased that our manuscript “Similar hemostatic responses to hypovolemia induced by hemorrhage and lower body negative pressure reveal a hyperfibrinolytic subset of non-human primate” received such a generally favorable response. 

Below are each of the points raised by the reviews and our response. We hope we have addressed to your satisfaction each point made and that our manuscript is now acceptable for publication in PLOS ONE.

Maryanne Herzig

We have adhered to PLOS ONE’s file naming requirements.

2. To comply with PLOS ONE submission guidelines for non-human primate studies, we request that you please provide the following information in the Methods section of your manuscript .

* Please provide details explaining the surgeries performed in this study. Thank you for providing a reference, we ask that you please include these details in this manuscript as well in the methods section or as supporting information.

This information is now included in S1_methods document and referenced in the methods section.

*Please describe the post-operative care received by the animals, including the frequency of monitoring and details of how you minimized animal suffering and distress.

This information is now included in S1_Methods document and referenced in the Methods section.

* Details of animal welfare (e.g., shelter, food, water, environmental enrichment)

This information is now included in S1_methods document and referenced in the methods section.

*The source of the animals used in this study, where they were bred and housed prior to this study.

This information is now included in S1_methods document and referenced in the methods section.

*Please explain the fate of the animals used in this study. If they were euthanized, please describe the method of sacrifice.

This information is now included in S1_methods document and referenced in the methods section.

3. We note that one or more of the authors are employed by a commercial company: Charles River Laboratories 

Within your Competing Interests Statement, please confirm that this commercial affiliation does not alter your adherence to all PLOS ONE policies on sharing data and materials by including the following statement: "This does not alter our adherence to PLOS ONE policies on sharing data and materials.” (as detailed online in our guide for authors Caution-http://journals.plos.org/plosone/s/competing-interests < Caution-http://journals.plos.org/plosone/s/competing-interests > ) . If this adherence statement is not accurate and there are restrictions on sharing ofdata and/or materials, please state these. Please note that we cannot proceed with consideration of your article until this information has been declared.

 While one of our authors (CB) is currently employed by Charles Rivers Laboratories, that affiliation was only listed as a professional courtesy. During the time of the study, she was employed by the Southwest National Primate Research Center, Texas Biomedical Institute as a veterinarian. After the study was completed, her contribution consisted of editing the written manuscript. At no point did Charles Rivers Laboratory participate or contribute to this study.

The Funding statement now reads: “This study was funded by the United States Army, Medical Research and Development Command. Charles Rivers Laboratories provided support in the form of salary for author CB during the editing of this manuscript but did not have any additional role in the study design, data collection and analysis, decision to publish, or preparation of the manuscript. All other authors have no other specific funding to declare.”

The Competing interests statement now reads: “While author C.B. is currently employed by Charles Rivers Laboratories, this commercial affiliation does not alter our adherence to PLOS ONE policies on sharing data and materials from this current manuscript. All other authors have no competing interests.”

The Author Contributions paragraph has been added to the document at Line 524. It reads: “Conceptualization: V.C., C.H-L., A.C., H.P., R.S.; Data Curation: C.F., M.Z., M.H., R.M., H.P.; Formal Analysis: M.Z., M.H., C.F., R.M., N.P., B.P., H.P.; Funding Acquisition: A.C., V.C., H.P.; Investigation: C.F., B.P., N.P., G.M., C.H-L., R.S., C.B., R.M., W.D., V.C.; Methodology: V.C., A.C., C.H-L., R.S., H.P., G.M.; Project Administration: C.H-L., H.P.,C.F., R.S., G.M., V.C.; Resources: R.S., C.B., G.M., H.P., A.C., C.H-L.; Writing – Original Draft Preparation: M.Z., M.H.; Writing – Review & Editing: M.Z., M.H., C.F., G.M., C.B., R.S., N.P., B.P., R.M., W.D., H.P., C.H-L., A.C., J.B., V.C.”

4. We note that you have included the phrase “data not shown” in your manuscript. Unfortunately, this does not meet our data sharing requirements. PLOS does not permit references to inaccessible data. We require that authors provide all relevant data within the paper, Supporting Information files, or in an acceptable, public repository.

”Data not shown” has been replaced with references to supporting data, S2_fig, Lines 370, 373, and 376.

Reviewers' comments:

Reviewer's Responses to Questions

 Nothing to address, all responses were affirmative

5. Review Comments to the Author

Reviewer #1: Global comments

It is well established that HEM causes hemodilution and LBNP causes hemoconcentration. In the present study it is shown, however, that TAFI activity and protein demonstrates hemodilution during LBNP and hemoconcentration during HEM. The authors should discuss on this discrepancy.

Thank you for catching this; despite multiple reviewers, it was misstated in the abstract. The information is correct in the body of the text and correct in the figures. The abstract now reads: “TAFI activity and protein, Factors V and VIII, vWF, Proteins C and S all demonstrated hemodilution during HEM and hemoconcentration during LBNP, whereas tissue plasminogen activator (tPA), plasmin/anti-plasmin complex, and plasminogen activator inhibitor-1 did not.” Line 43

Sustained response of baboons to blood removal 4 weeks previously was observed. Authors should discuss on possible physiological/biochemical reasons.

It was surprising to find sustained changes 28 days after a blood loss of less than 1h duration considering the blood was re-infused. Whether this is a hemodilution effect of the introduction of the anti-coagulant with the blood, a citrate effect of anti-coagulant, an effect of sedation or a hemostatic response to rapid blood loss despite re-transfusion is unknown. Ketamine has a known effect of increasing platelets 15 days after infusion (Colic, 2018), why this would extend to hemostatic proteins such as Protein C, TAT or TAFI activity, all related to thrombin regulation, is unclear.

To emphasize the possibility of ketamine sedation on sustained effects, the phrase “performed under ketamine sedation” has been added to line 196 ….previous HEM trial performed under ketamine sedation.

The authors state that blood samples for plasma analyses were briefly stored in wet ice before centrifugation (lines 138-139). However, storage at low temperatures can cause platelet activation. Can authors exclude that platelet activation occurred in their study? 

We cannot exclude that platelet activation occurred in the samples destined for plasma protein analysis, thus the critical comparison is to BL values of samples treated in the same way. However analyses of platelets by flow, thromboelastography and platelet aggregation were all performed on samples not placed on ice.

 The authors state that platelet aggregation measurements were performed by using heparin tubes (lines 160-161). However, platelet aggregation measurement are usually performed in citrated blood samples (see Cvirn et al. (2012) Coagulation changes during presyncope and recovery. PLOS ONE 7 (8), e42221). Can authors comment on this?

While platelet aggregation is often performed on citrated tubes, a study by Dugan et al found non-human primates with more reliable values using heparin tubes: “Assessment of Multiplate® platelet aggregometry using Citrate, Heparin or Hirudin in Rhesus macaques”. Greg Dugan, Lisa O’Donnell, David B. Hanbury, J. Mark Cline, and David L. Caudell, Platelets. 2015; 26(8): 730–735. This reference is now included in the Methods section. Line 158.

Specific comments

1. The title is a bit misleading. The study was not designed to identify or analyze the hyperfibrinolysis subset of non - human primates. Revision of the title should be considered, e.g., „Similarities in hemostatic response to LBNP and controlled hemorrhage in baboons – a crossover study“.

We respectfully disagree with the reviewer. The study may not have been designed to identify the hyperfibrinolysis subset of non-human primates, but we felt this important finding should be highlighted in the title.

Changing the title to: “Similarities in hemostatic response to LBNP and controlled hemorrhage in baboon-a crossover study” would be misleading as this was not a cross-over design. 

2. Line 70/ Line 72: a newer reference about LBNP should be included (Goswami et al. 2018. Lower body negative pressure: Physiological effects, applications, and implementation. Physiol Rev. 2019; 99(1): 807-851)

Reference added- Line 68

3. Line 77: Newer references about LBNP and coagulation should be included (Cvirn et al. 2019 Coagulation changes induced by lower-body negative pressure in men and women. J Appl Physiol (1985). 2019; 126(5):1214-1222).

Reference added-Line 74

4. There is no indication of the study design.

Study design added in supplementary figure: S1_Fig, Line 98

5. Sample size: Were sample size calculations done? If yes, on what basis?

As this was a pilot/preliminary study done in an expensive large animal model, no sample size calculations were done. Twenty five animals were allotted to this arm of the study, but the data was robust with complete data for 14 animals and the study was curtailed due to financial considerations. Using a repeated measures design allowed each animal to be compared to itself; comparison to independent groups would have required a larger sample size. 

6. In the study is acknowledged that the carry-over effect exists that extends over the washout period; however, it is incorrect to ignore the design of the study and just perform a comparison of treatments (test conditions). The possibility of a period effect should be tested – to compare the differences between the periods in the group. Additionally, the possibility of the treatment period interaction also exists (see: Altman DG. Clinical trials. In: Practical statistics for medical research. London: Chapman & Hall; 1999. p. 467–71). In the study, that would not be possible, since subject allocation to one or the other test condition (HEM or LBNP) was not random (randomization was not achieved). The authors should discuss this.

The possibility of an order effect is discussed in the limitations wherein the importance of the non-random order for matched LBNP pulse pressure and CVP to Hem are emphasized. 

Whether this is a hemodilution effect of the introduction of the anti-coagulant with the blood, a citrate effect of anti-coagulant, an effect of sedation or a hemostatic response to rapid blood loss despite re-transfusion is unknown. Ketamine has a known effect of increasing platelets 15 days after infusion (Colic, 2018), why this would extend to hemostatic proteins such as Protein C, TAT or TAFI activity, all related to thrombin regulation, is unclear.

This above statement has been added to the discussion line 425

7. The limitations sections should be expanded so that it takes into account the above outlined concerns.

The limitations sections now include the phrase: “While differences between HEM and LBNP hemostatic responses could be explained by an order effect, no such differences were observed.” line 491

8. Lines 205-206: It is stated here that a low correlation existed between RBC and platelet count at HEM BL (Fig. 1; r=0.05). This correlation, however, is not presented in Fig. 1.

Added to supplementary S2_Fig- Lines 370, 373, and 376.

Reviewer #2: The authors describe, characterize and compare hemostatic profiles of two techniques of modeling hypovolemia in non-human primates; either hemorrhage or lower body negative pressure. The authors have a well written manuscript. the coagulation response of each technique provides important information to researchers in this area. Studies using non-human primates are essential to promote improvements in care for the injured patient. The authors findings are novel and robust. There conclusions are appropriate and limitations well described.

Thank you.

Supporting information begins at line 595 after references and is as follows:

Supporting Information.

S1_Methods. Animal Methods

S1_Fig. Test design and Study Protocol

S2_Fig. Post Hoc analysis of correlations for baboons grouped as low-lysis (LY30 <2%) or high-lysis (LY30 >2) for tissue-type plasminogen activator (tPA) vs. plasmin-�2-antiplasmin complex (PAP) and vs. von Willebrand factor (vWF); clot lysis at 30 min (LY30) vs. D-dimer; and red blood cell count (RBC) vs. platelet count (PLT) at both BL and MAX hypovolemia.

---

## [Editor Report · Decision Letter 1]

4 Jun 2020

Similar hemostatic responses to hypovolemia induced by hemorrhage and lower body negative pressure reveal a hyperfibrinolytic subset of non-human primate

PONE-D-19-35990R1

Dear Dr. Herzig,

We are pleased to inform you that your manuscript has been judged scientifically suitable for publication and will be formally accepted for publication once it complies with all outstanding technical requirements.

With kind regards,

Nandu Goswami, MBBS, PhD

Academic Editor

PLOS ONE

Additional Editor Comments (optional):

All the concerns of reviewer 1 have been addressed in your revised submission.
---

## [Editor Report · Acceptance letter]

9 Jun 2020

PONE-D-19-35990R1 

Similar hemostatic responses to hypovolemia induced by hemorrhage and lower body negative pressure reveal a hyperfibrinolytic subset of non-human primates 

Dear Dr. Herzig:

I'm pleased to inform you that your manuscript has been deemed suitable for publication in PLOS ONE. Congratulations! Your manuscript is now with our production department. 

Kind regards, 

on behalf of

Dr. Nandu Goswami 

Academic Editor

PLOS ONE